# High spatial resolution nanoslit SERS for single-molecule nucleobase sensing

Chang Chen [1,2], Yi Li [1,3], Sarp Kerman[1,2], Pieter Neutens[1,2], Kherim Willems [1,4], Sven Cornelissen[1,3], Liesbet Lagae[1,2], Tim Stakenborg[1] & Pol Van Dorpe[1,2]

Solid-state nanopores promise a scalable platform for single-molecule DNA analysis. Direct, real-time identification of nucleobases in DNA strands is still limited by the sensitivity and the spatial resolution of established ionic sensing strategies. Here, we study a different but promising strategy based on optical spectroscopy. We use an optically engineered elongated nanopore structure, a plasmonic nanoslit, to locally enable single-molecule surface enhanced Raman spectroscopy (SERS). Combining SERS with nanopore fluidics facilitates both the electrokinetic capture of DNA analytes and their local identification through direct Raman spectroscopic fingerprinting of four nucleobases. By studying the stochastic fluctuation process of DNA analytes that are temporarily adsorbed inside the pores, we have observed asynchronous spectroscopic behavior of different nucleobases, both individual and incorporated in DNA strands. These results provide evidences for the single-molecule sensitivity and the sub-nanometer spatial resolution of plasmonic nanoslit SERS.

[1] imec, Kapeldreef 75, 3001 Leuven, Belgium. [2] Department of Physics and Astronomy, KU Leuven, 3001 Leuven, Belgium. [3] Department of Electrical Engineering, KU Leuven, 3001 Leuven, Belgium. [4] Department of Chemistry, KU Leuven, 3001 Leuven, Belgium. These authors contributed equally: Chang Chen, Yi Li. Correspondence and requests for materials should be addressed to C.C. (email: chang.chen@imec.be)

The importance of DNA sequencing is steadily increasing for biomedical research and it is entering clinical diagnostics[1,2]. The recently launched and rapidly growing field of nanopore-based DNA sequencing provides a promising way for portable, easy, and low-cost purposes[3–6]. Nanopore sequencing strategies are predominantly based on monitoring fluctuations of the ionic current flowing longitudinally through the pore, with some initial reports of transversal detection schemes based on quantum tunneling or transistor-like designs[7–10]. Such nanopores can realize label-free detection of DNA through resolving the intimate interactions between the ionic fluids and molecules mechanically confined inside the pore[11,12]. Using an engineered biological pore, it is possible to distinguish the individual nucleobases by the specific amplitude variations of the ionic current signals[13–15]. The accuracy of identification can be further improved by adding polymer labels to the nucleotides to amplify the current variations[16]. Meanwhile, alternative or complementary sensing strategies, such as laser spectroscopies including fluorescence and surface enhanced Raman spectroscopy (SERS), have recently been proposed for nanopore sequencing[17–19].

To enable DNA sequencing, we should address three fundamental challenges: (1) an accurate identification mechanism, (2) single-molecule sensitivity, and (3) single-molecule spatial resolution. Combining SERS with nanopore sensing in an aqueous environment is a promising strategy for single-molecule identification. Already two decades ago, it was demonstrated that SERS could be employed to identify single molecules, and subsequently it has been proposed for DNA sequencing[20–22]. Because SERS provides spectroscopic information related to the chemical structure of analytes, it has an extraordinary advantage for accurate identification[23]. Pioneering studies from different groups have demonstrated the capability of identifying DNA or RNA fragments and even individual nucleobases by SERS[18,24–26]. Moreover, recent theoretical and experimental studies have demonstrated that the spatial resolution of SERS approaches the sub-nanometer level in dry conditions[27–29]. In a recent numerical

report, the feasibility of nanopore SERS for sequencing was discussed, and clear advantages with respect to both molecular control and identification were identified[30]. In addition, as an optical spectroscopy technique, SERS does not interfere with ionic detection, allowing for a multi-modal sensing strategy.

In this report, we experimentally study the feasibility of combining SERS with nanopore fluidics for addressing the three fundamental challenges related to DNA sensing and even sequencing. By using elongated plasmonic nanoslits, rather than the conventional round plasmonic nanopores, we first establish a spectroscopic library for accurately identifying the four nucleotides. Based on voltage modulation, we can further realize a temporal and stochastic fluctuations of molecules adsorbed inside the hot spots of the nanoslit, which needs to be carefully matched with the slow sampling speed of SERS. When the setup is ready for real-time molecular sensing in fluid, we study the fluctuation process of adsorbed adenine isotopologues following the bi-analyte SERS (BiASERS) strategy[31]. And we can observe single-molecule sensing events. By studying the fluctuations of adsorbed DNA oligonucleotides, we can identify adjacent bases incorporated in DNA strands, indicating the sub-nanometer spatial resolution of the nanoslit SERS. Although we cannot study these fast translocation events of DNA molecules yet, the discussed features of nanoslit SERS can still be interesting for single-molecule DNA sensing in a label-free and real-time way.

## Results

**Plasmonic nanoslit devices.** The plasmonic nanoslit devices in use were specifically designed and fabricated for real-time, localized SERS measurements in liquid conditions[32,33]. As shown in Fig. 1a and Supplementary Fig. 1, a typical plasmonic device consists of a sub-10 nm wide and 1 μm long nanoslit, a cavity and two Bragg-mirror gratings in a same freestanding membrane. The nanoslit is located at the bottom of the cavity, while the gratings are aside the cavity. Optically, the cavity can efficiently couple the incident light into surface plasmon polaritons (SPPs) and guide

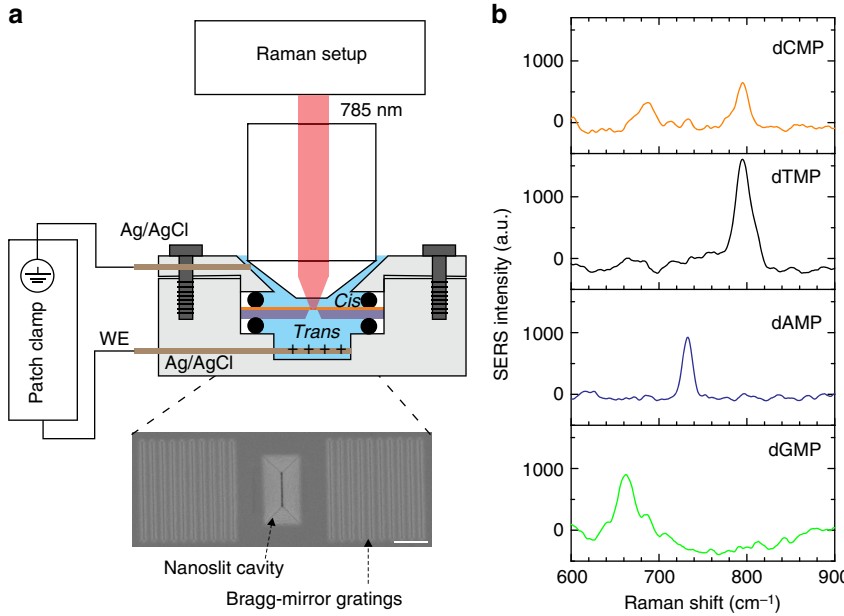

**Fig. 1** Plasmonic nanoslit SERS. **a** Schematic representation of the setup for nanoslit SERS. The nanoslit chip is sealed in a flow cell which separates the electrolyte solution into two compartments. The top chamber can accommodate a water-immersion objective lens. A 785 nm laser with 8 mW is focused on the gold nanoslit. Axon patch 200B amplifier is used to apply the transmembrane voltages and monitor the ionic currents between two Ag/AgCl electrodes. The inset shows a top-view SEM image of the nanoslit structure, consisting of an inverted prism nanoslit cavity with Bragg-mirror gratings. The scale bar is 1 μm. **b** SERS spectra of four DNA nucleotides. Each spectrum was averaged from 100 spectra taken from the specific nucleotide solution of $1 \times 10^{-3}$ M in 10 mM KNO$_3$. The acquisition time was 0.5 s and the applied voltage was +0.4 V

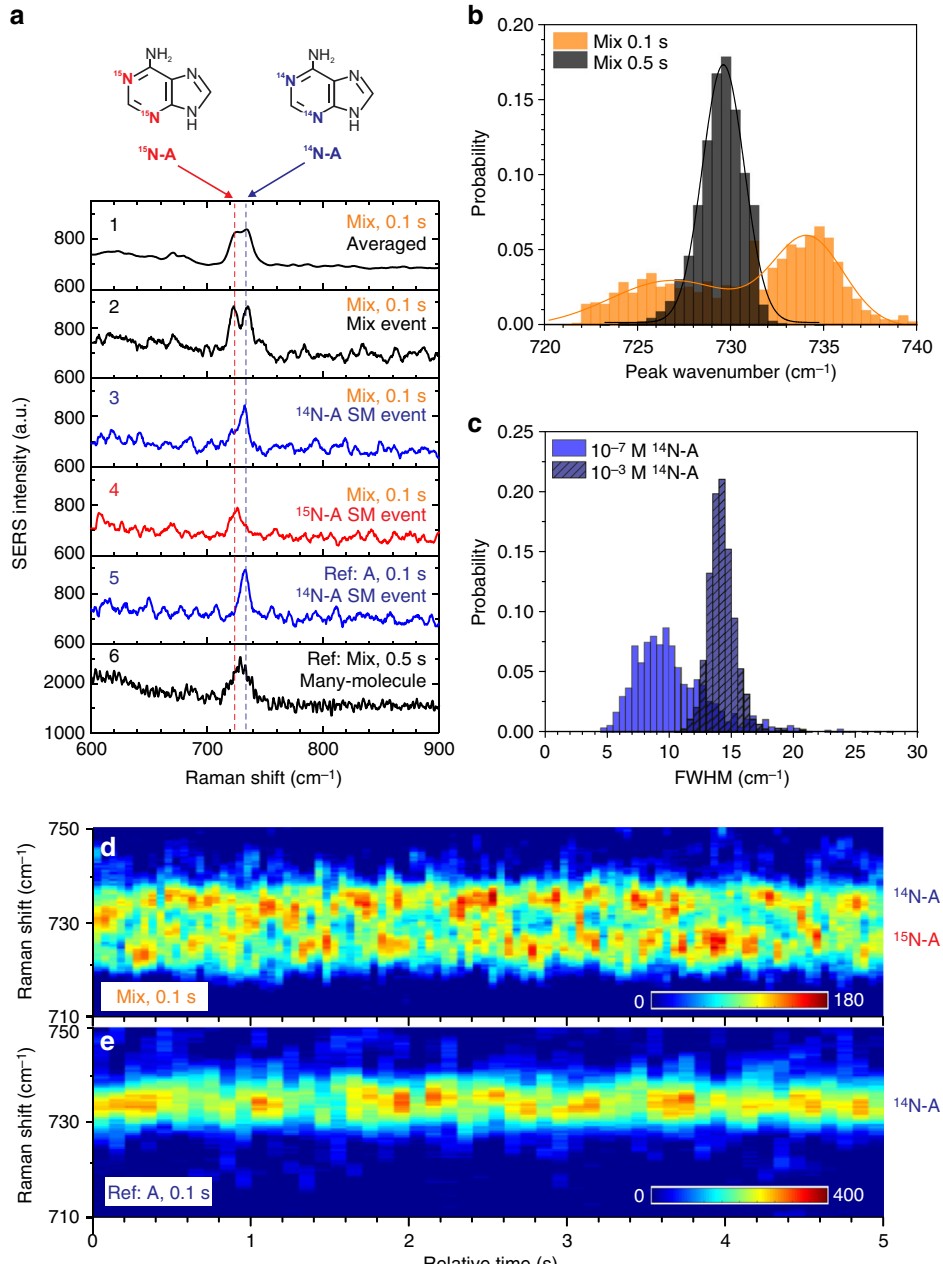

**Fig. 2** Single-molecule sensitivity. **a** Representative SERS spectra from a real-time BiASERS of a mixture solution of $1 \times 10^{-7}$ M $^{14}$N-adenine and $^{15}$N-adenine in 10 mM KNO$_3$. Spectrum 1 is averaged from all spectra, spectrum 2 refers to a mixed event of both adenines, spectrum 3 refers to a single-molecule event of $^{14}$N-adenine, and spectrum 4 refers to a single-molecule event of $^{15}$N-adenine. All spectra were taken at +0.5 V and 0.1 s. The two reference spectra 5 and 6, one was taken from a different solution with only $^{14}$N-adenine ($10^{-7}$ M in 10 mM KNO$_3$), acquired for 0.1 s and the other was taken from the same mixture solution but with a prolonged acquisition of 0.5 s. **b** The effect of acquisition time on the distribution of the peak wavenumber. Prolonging the acquisition time from 0.1 to 0.5 s, the dual-peak distribution of the peak wavenumber becomes a single-peak distribution (2500 spectra for each). **c** Concentration effect on the distribution of the FWHM. FWHM of Raman bands from samples (1000 spectra taken at 0.1 s) at a lower concentration of $10^{-7}$ M is smaller than that from a higher concentration sample of $10^{-3}$ M. Both the spectral merging and broadening of bands indicate the transition of single-molecule to many-molecule sensing. **d, e** The contour maps of SERS of a typical asynchronous blinking of the mixed isotopic adenines (**d**) and a typical fluctuation of the single adenine (**e**) recorded in 5 s, respectively in single-molecule sensing. The excitation (785 nm, 8 mW), the applied voltage (+0.5 V) and the nanoslit in use were the same for all experiments

SPPs into the slit. The plasmonic Bragg-mirror gratings can reflect SPPs back into the slit-cavity to further strengthen the optical field. Only inside the slit, the plasmonic gap mode generates a hot spot region with a strong and broadband electromagnetic field enhancement for SERS. In addition, this design of plasmonic nanoslit is compatible with complementary metal–oxide–semiconductor (CMOS) fabrication processes,

enabling wafer-level mass-manufacturing (see Supplementary Fig. 1b)[33,34]. To reduce the capacitance of the membrane and prevent potential chemical reactions occurring at the interface, we have deposited a 50 nm SiO$_2$ layer at the backside (Si side)[35]. The asymmetric plasmonic nanoslit cavity is facing up toward the top *cis*-chamber, and the working electrode (WE) is in the bottom *trans*-chamber. Upon application of a positive bias voltage, the

nanoslit provides the only path, through which negative-charged DNA molecules translocate from the *cis* to the *trans*-chamber.

In contrary to reported plasmonic nanopores, such as the round or square nanopores with plasmonic antennas (e.g., bow tie antenna) and the Fabry–Pérot nanopores, that only have a single hot spot region for sensing[33,36–38], the nanoslit described here, results in a more complex local hot spot pattern. The inherent roughness of the gold surface creates a locally fluctuating gap width, leading to the possible formation of multiple, random hot spots along the long axis of the nanoslit. During the experiment, the analyte molecules can diffuse in and out of the hot spots. The slow sampling rate of the Raman setup, however, only allows to study DNA molecules temporally adsorbed at the hot spots. Nevertheless, the broadband and large field enhancement in the nanoslit, and the intrinsic alignment to the translocating nanofluid, make the nanoslit an interesting model system for understanding the properties of single-molecule spectroscopic nanopore sensing.

**Nucleotide identification**. We first investigated the ability of nanoslit SERS for identifying different nucleotides. For this, we constructed a SERS library of nucleotides by measuring each nucleotide sample at a high concentration of $1 \times 10^{-3}$ M in 10 mM $KNO_3$. Different plasmonic nanoslits with similar gap sizes (<10 nm) were used. We recorded 100 continuous spectra from every nucleotide sample. Figure 1b shows the averaged spectra of these four nucleotides. A spectroscopic range of 600–900 cm$^{-1}$ is chosen for the bulk of the data analysis in this work. In this range, there are only the major characteristic Raman bands attributed to the strong breathing modes of the six-membered aromatic ring of the nucleotides. These unique Raman bands shown in Fig. 1b, namely 661 cm$^{-1}$ for dGMP, 732 cm$^{-1}$ for dAMP, 795 cm$^{-1}$ for dTMP, and 796 cm$^{-1}$ for dCMP are in line with SERS measurements taken using other substrates[39,40]. In the histograms of the peak wavenumber and the full width at half maximum (FWHM) of these bands (see Supplementary Note 1 and Supplementary Fig. 2), it is shown that most FWHMs are between 13 and 20 cm$^{-1}$, and the peak wavenumbers of nucleotides are very distinct, except those of dTMP and dCMP are quite similar. To accurately identify dTMP and dCMP, the involvement of other bands like the one at 687 cm$^{-1}$ for dCMP (deformation mode of the aromatic ring) or other spectroscopic features appearing at a higher wavenumber (see Supplementary Fig. 2) is required[40]. It should be noted that to improve the capture and adsorption efficiencies of the nanoslit, we applied a positive bias voltage (e.g., +0.4 V) to drive the motion (translocation and adsorption) of the DNA samples. As discussed in Supplementary Note 2 and shown in Supplementary Fig. 3, the SERS signal of dAMP becomes much stronger with a positive bias voltage. At neutral or negative biases, the observed SERS signals are much weaker or even undetectable. This voltage modulation is effective for all five nanoslits used in this work, as well as other kind of nucleotides. Without the presence of the analyte molecules, we have no SERS signals under the applied bias voltage (see Supplementary Fig. 4).

**Single-molecule sensitivity**. To study the single-molecule sensitivity of nanoslit SERS, we followed the strategy of BiASERS developed by Le Ru and et al.[41]. In general, BiASERS relies on observations of independent combinations of different analytes appearing in the hot spots present in the focal volume of the Raman setup[42]. At the single-molecule level, such combinations are shown as asynchronous blinking of spectroscopic features of the two different analyte molecules. A statistical analysis of the data can prove the independence of these events, while the single-molecule sensitivity can be demonstrated when a selection of the

obtained spectra can be attributed exclusively to one of the mixed analyte molecules, by using its specific spectroscopic fingerprint. In nanoslit SERS, since the hot spots were spatially and temporally fixed, the different combinations can only rely on the fluctuations of the molecules at the hot spots of the slit.

The most convincing BiASERS method is using two analyte molecules having nearly identical affinities to the Au surface. We can realize this by using two chemically similar but spectrally different molecules, such as a pair of isotopologues[31,43]. Here, we used equimolar concentrations of a pair of isotopic adenines of $^{14}$N-A (common) and $^{15}$N-A(rare), which have the same affinity to the gold surface[24,44]. In this real-time BiASERS experiment of the adenine isotopologues, we monitored the fluctuations of spectroscopic features: the intensity, FWHM, and the peak wavenumber of the analyte Raman bands. These spectroscopic fluctuations can be originated from displacements of different molecules and/or spatial fluctuations of the same molecules in and out the hot spots of the nanoslit. Since the Raman bands of isotopic adenines are close in frequency, in many-molecule sensing, many independent spectroscopic fluctuations are superimposed to inhomogeneously broaden the resultant Raman bands. While in single-molecule sensing, distinct bands from isotopic variations can still be clearly resolved within this inhomogeneous broadened range. This forms two spectroscopic features of the single-molecule sensing: a narrower FWHM and a shifting peak wavenumber within the inhomogeneous broadened range.

We studied these spectroscopic features at different experimental configurations: single and mixed adenine isotopologues, low and high concentrations, and fast and slow acquisitions. Single-molecule sensing has a larger likelihood at a low analyte concentration of $10^{-7}$ M, as optimized in our nanoslit SERS. At this concentration, we can observe asymmetric spectroscopic superposition and the band splitting within the inhomogeneous broadening of the Raman band at ~730 cm$^{-1}$, representing the characteristic features of single-molecule sensing[31,45]. Fig. 2 shows the SERS results of a mixture solution and a single-analyte solution (reference) at the same low concentration of $1 \times 10^{-7}$ M in 10 mM $KNO_3$. From the mixture solution, we recorded 2500 spectra continuously at 0.1 s acquisition time per spectrum for our statistical analysis. Details of all spectra are shown in Supplementary Note 3.

In Fig. 2a, we show several typical spectra to explain BiASERS and the inhomogeneous broadening behavior. Spectrum No. 1 is the average of all 2500 spectra from the mixture sample. It shows a broad band induced by inhomogeneous broadening from many spectroscopic fluctuations. These 2500 spectra contain several spectra with distinct features. For instance, spectrum No. 2 exhibits a double band with similar amplitudes, within the broader range shown in No. 1. From separate ensemble measurements of $^{14}$N-A and $^{15}$N-A (see Supplementary Fig. 5 and 6), the left band at 723 cm$^{-1}$ can be attributed to $^{15}$N-A, and the right band at 734 cm$^{-1}$ can be attributed to $^{14}$N-A. To further verify the single-molecule sensitivity, we identified several spectra that can likely be assigned to true single-molecule events. For instance, spectrum No. 3 has only a single peak at 733 cm$^{-1}$, matching with spectra taken from a reference sample of isotopically pure $^{14}$N-A at the same concentration and the same acquisition times (see Fig. 2a, spectrum No. 5). While spectrum No. 4 has a single band at 726 cm$^{-1}$, representing a single-molecule event of $^{15}$N-A. As shown in Supplementary Table 2, these single-band spectra attributed to a single isotopic adenine were few. Mostly, the amplitude of the two Raman bands of isotopologues varied randomly due to the stochastic fluctuations of molecules. As shown in Fig. 2d, in an exemplary 5 s real-time measurement, the dominating band was switching frequently

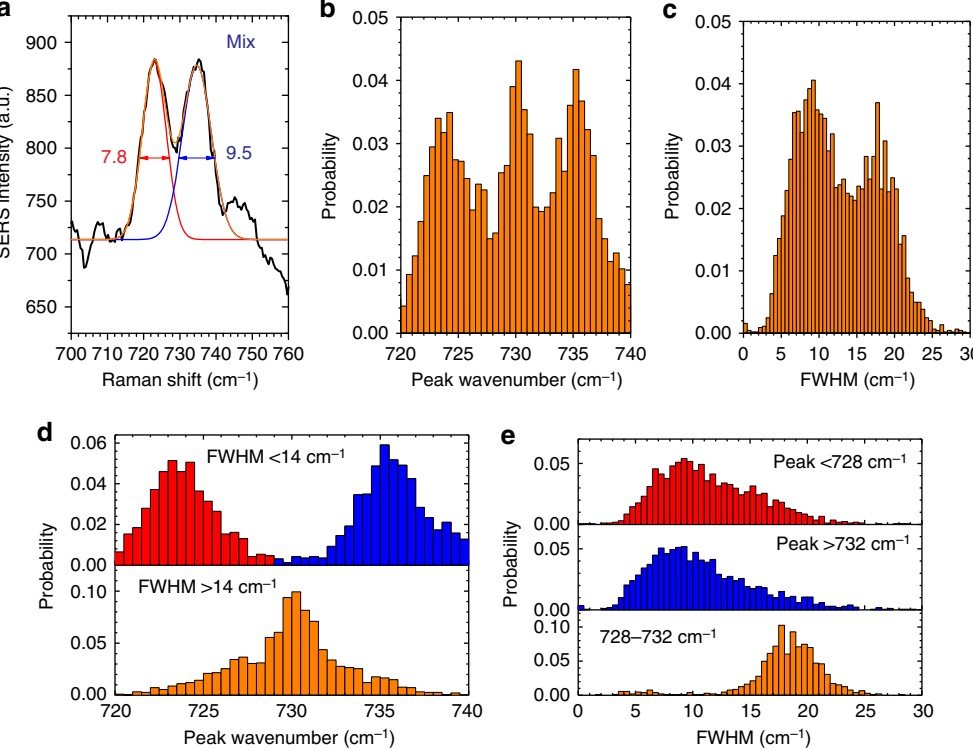

**Fig. 3** Deconvolution of Raman bands obtained in BiASERS. **a** An example of the deconvolution algorithm. For a mixed event, the deconvolution in the spectroscopic range of 710–750 $cm^{-1}$ can find two peaks: the red one represents the $^{15}N$-adenine with the FWHM of 7.8 $cm^{-1}$ and the peak at 723 $cm^{-1}$, while the blue one represents $^{14}N$-adenine with the FWHM of 9.5 $cm^{-1}$ and the peak at 734 $cm^{-1}$. The orange line is the merged curve of two fittings, which perfectly fits the experimental spectrum (black). **b**, **c** Histograms of the peak wavenumber and the FWHM of all deconvolved Raman bands. **d** Extracted histograms of the peak wavenumber filtered by the valley value of the FWHM distribution (14 $cm^{-1}$). **e** Extracted histograms of the FWHM filtered by the valley values of the peak wavenumber distribution (728 and 732 $cm^{-1}$). The correlation between narrower FWHMs and distinct peak wavenumbers indicates the single-molecule sensing of each adenine isotopologues. 2500 SERS spectra from a real-time BiASERS of a mixture solution of $1 \times 10^{-7}$ M $^{14}N$-adenine and $^{15}N$-adenine in 10 mM $KNO_3$, were deconvoluted and obtained 4410 fitted Raman bands

between the two wavenumbers assigned to $^{15}N$-A and $^{14}N$-A, respectively. In the reference experiment of $^{14}N$-A (see Fig. 2e), we did not observe the switching of the peak wavenumber, but only the intensity fluctuation of the Raman band of $^{14}N$-A. Such formed symmetric double bands or asymmetric broadening bands indicated that only single or few molecules were involved in nanoslit SERS. As a control experiment, we prolonged the acquisition time from 0.1 to 0.5 s to involve more spectroscopic fluctuations (see Supplementary Fig. 7). As expected, we only observed stable, single band spectra (see Fig. 2a, spectrum No. 6). To study a solid case of many-molecule sensing, we accomplished another control experiment on a mixture at a 100-fold higher concentration of $1 \times 10^{-5}$ M, and we did not observe any double bands or asymmetric broadening behavior, but only a single, broad band, which we assigned to an ensemble of both isotopic adenines (see Supplementary Fig. 8).

**Single-molecule statistics in temporal BiASERS.** We also statistically analyzed the fluctuations of bands to further validate the single-molecule sensitivity of nanoslit SERS. We chose the peak wavenumber and the FWHM to represent different isotopic variations. In Fig. 2b, it is shown that the distribution of peak wavenumbers of the strongest bands highly depends on the chosen acquisition time. For the "fast" acquisition of 0.1 s per spectrum, there is a clear dual-peak distribution (orange) of the peak wavenumber (defined by the local max algorithm) at 726 and 734 $cm^{-1}$, respectively representing the $^{15}N$-A and $^{14}N$-A Raman bands. This dual-peak distribution of bands can directly represent the inhomogeneous broadening effect caused by the

isotopic variations at the single (or few) molecule level[45]. When involving more molecules and fluctuations, such as prolonging the acquisition to 0.5 s, there is only a single-peak distribution of the peak wavenumber centered at ~729 $cm^{-1}$, in between the $^{14}N$-A and $^{15}N$-A peak positions (see Fig. 2b). This shows that integrating over more isotopic fluctuations results in a single, merged Raman band. To corroborate the dual peak result, we performed a reference experiment of a single-analyte solution of $^{14}N$-A at the same low concentration of $1 \times 10^{-7}$ M. And we could only find a single-peak distribution centered at 733 $cm^{-1}$ (see Supplementary Fig. 9b). The FWHM is another characteristic spectroscopic feature in single-molecule experiments. Because spectral broadening of the Raman bands usually occurs in an ensemble SERS measurement[45], we can expect narrower FWHMs of Raman bands in a single molecule measurement. As shown in Fig. 2c, the FWHM of the Raman bands at a low concentration of $10^{-7}$ M is ~9 $cm^{-1}$, which is substantially narrower than the FWHM of 14 $cm^{-1}$ obtained from a higher concentration $^{14}N$-A solution of $10^{-3}$ M. In other control experiments of many-molecule sensing, we have also observed wider FWHMs (see Supplementary Fig. 7 and 8).

To better study the features of the single-molecule features in BiASERS, we used a deconvolution algorithm to fit the different bands in all individual 2500 spectra. We first used the local maximum algorithm to find the initial wavenumbers and the number of peaks for Gaussian fitting. An example of the process is shown in Fig. 3a. Two peaks were found by the local maximum algorithm and fitted by the Gaussian fitting algorithm. They appear at 726 and 735 $cm^{-1}$, with narrow FWHMs of 7.8 and 9.5

cm$^{-1}$, respectively corresponding to $^{15}$N-A and $^{14}$N-A Raman bands. We further deconvolved all 2500 spectra in the range of 700–760 cm$^{-1}$ and listed the peak features of all fitted 4410 bands in Fig. 3. Different to the dual-peak histogram shown in Fig. 2b, we obtained a triple-peak distribution of peak wavenumbers in Fig. 3b. These three peaks at ~724, 730, and 736 cm$^{-1}$ are related to $^{15}$N-A, mixture, $^{14}$N-A, respectively. With deconvolution, we could find more hidden Raman bands. In Fig. 3c, we can find a dual-peak distribution of FWHMs, representing the distributions of single/few molecules (~9 cm$^{-1}$) and many molecules (~18 cm$^{-1}$). When we used the valley value of the FWHM distribution, 14 cm$^{-1}$, to filter peak wavenumbers, we may extract the features of single-molecule and many-molecule sensing of adenines. As shown in Fig. 3d, when FWHMs are <14 cm$^{-1}$, these Raman bands can be clearly assigned to either $^{15}$N-A (red part) or $^{14}$N-A (blue part). This indicates the single/few molecule sensing. When FWHMs are >14 cm$^{-1}$, these Raman bands are assigned to the mixture of $^{15}$N-A and $^{14}$N-A (orange part). This indicates more molecule involved in sensing. We can further use the same strategy for analyzing the distribution of FWHMs. FWHMs were filtered by the two valley values, 728 and 732 cm$^{-1}$ of the peak wavenumber distribution. As shown in Fig. 3e, when Raman bands appear at <728 cm$^{-1}$ (refer to $^{15}$N-A) or >732 cm$^{-1}$ (refer to $^{14}$N-A), their median FWHMs are ~9 cm$^{-1}$. When the bands appear at 728–732 cm$^{-1}$, the median FWHM is ~18 cm$^{-1}$. The correlation of narrower FWHMs and distinct peak wavenumbers indicates the single-molecule sensing of each adenine isotopologue. While the correlation of wider FWHMs and the merged peak wavenumber shows the many-molecule sensing of the mixed isotopologues. Due to the stochastic fluctuations of molecules, many-molecule events were

unavoidable. But they can be distinguished from single-molecule events. We summarized the composition of contributions of adenine isotopologues in Supplementary Table 2. As indicated, 1.5% of the spectra are true single-molecule events with a clear single-band of either $^{15}$N-A or $^{14}$N-A. 63.4 % spectra can be attributed to few-molecule events with a clear symmetric or asymmetric double-band of $^{15}$N-A and $^{14}$N-A. The remaining spectra are likely many-molecule events. To investigate whether the observation of $^{15}$N-A and $^{14}$N-A is independent of each other, we further modeled the probabilities of transitions between the three states dominated by $^{15}$N-A, $^{14}$N-A, and their mixture in a Markov process (see Supplementary Note 4 and Fig. 11). Clearly, we obtain different transition probabilities, and none of them are close to zero. Each adenine isotopologue fluctuated strongly and independently, providing an evidence for observing independent single-molecule events.

These BiASERS experiments using isotopic adenines unambiguously demonstrate the single-molecule sensitivity of nanoslit SERS. It must be noted that the obtained results were very sensitive to the exact experimental setting. In our screening experiments, we have varied many parameters: analyte concentrations between 10$^{-9}$ and 10$^{-3}$ M, the bias voltage between $-1$ and $+1$ V, and the acquisition time between 0.001 and 1 s. The current setting of 10$^{-7}$ M, $+0.5$ V, and 0.1 s turned out to be the sweet spot to obtain single-molecule results. Diluting the concentration 10 times to 10$^{-8}$ M and using 10 times longer acquisition of 1 s did not show any SERS events within the duration of the experiment. Within this small working window, results were reproducible and single-molecule sensitivity could be demonstrated using other nanoslit devices with similar morphologies. However, different types of plasmonic nanopores, may

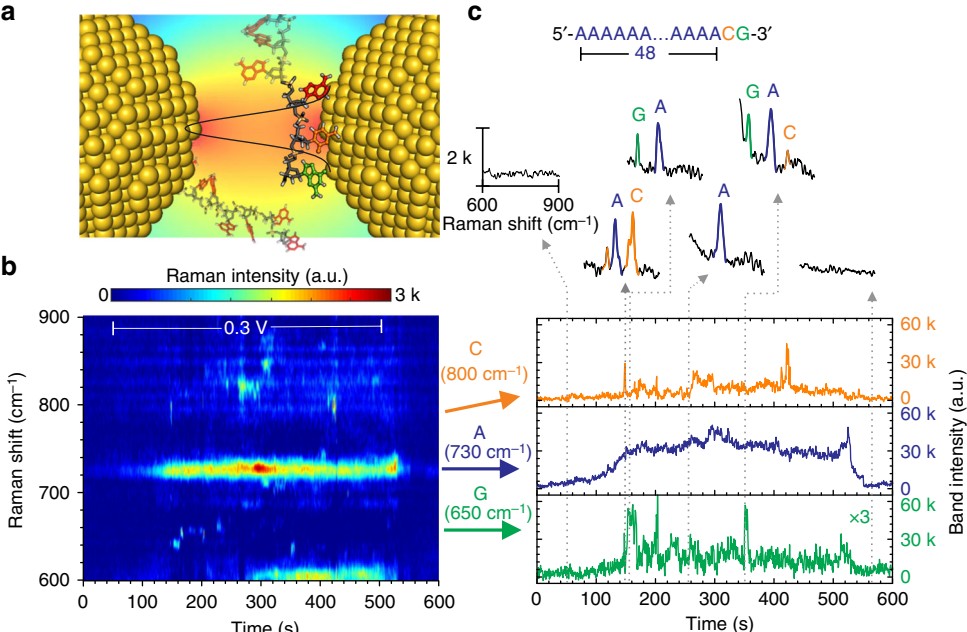

**Fig. 4** Single-molecule spatial resolution. **a** Schematic representation of temporarily adsorbed ss-DNA inside the nanoslit. The background represents the localization of the optical field inside the nanoslit. The black curve represents the longitudinal profile of the SERS amplification near the gold surface, which provides a local sensing at the sub-nanometer scale. The few DNA bases present inside this black curve contribute to SERS signals, while the others (semi-transparent) outside this region do not. Objects are not in the same scale. **b** The temporal contour map of nanoslit SERS of a 50-mers ss-DNA oligonucleotides 5′-poly(dA)$_{48}$dCdG-3′ (1 × 10$^{-8}$ M in a 10 mM KNO$_3$ solution). A voltage of +0.3 V was applied between 50 and 550 s. Each spectrum was acquired for 0.5 s. The different colored dash-line frames indicate the characteristic Raman bands of G (green), A (blue), and C (orange), respectively. **c** The temporal traces of the band intensity of nucleobases marked in **b**. The persistent signal in the trace of A indicates the voltage driven adsorption of ss-DNA inside the nanoslit. While the spikes in traces of G and C indicate asynchronous molecular identifications of distinct nucleobases during the stochastic fluctuations of the DNA strand. The insets show 6 representative spectra. The asynchronism of spikes from adjacent nucleobases indicates the sensing at a sub-nanometer resolution

require a different set of experimental parameters to enable single-molecule SERS.

In principle, nanoslit SERS can detect other bases at the single-molecule too and hence should not be restricted to isotopologues. As shown in Fig. 1b, for a similar experimental configuration, the SERS intensities of the characteristic bands of the other nucleobases are comparable to adenine. To prove it, we have measured dGMP at the same low concentration of $10^{-7}$ M, +0.5 V voltage, and 0.2 s acquisition. As shown in Supplementary Fig. 10, we observed the same characteristic features of single-molecule sensing, namely narrower FWHMs and a broader distribution of the peak wavenumbers of the recorded Raman bands than these Raman features obtained in many-molecule sensing (see Supplementary Fig. 2). In another experiment, we measured a mixed solution of all four nucleotides at $10^{-5}$ M (see Supplementary Note 5). As shown in Supplementary Fig. 12, step-like blinking signals of dCMP can be observed in the presence of relative stable signals of dGMP, while the other two nucleotides were not detected. Although the different affinity of the four nucleotides to the gold surface complicates the interpretation of BiASERS, we could still observe some true single-molecule events in BiASERS of this mixture solution. The FWHMs of these blinking signals of dCMP are narrower than that obtained in many-molecule sensing. Likely this can be explained by the combination of many adsorbed dGMP molecules and single dCMP molecules entering the hot spot(s).

**Sub-nanometer spatial resolution**. The third fundamental challenge is the single-molecule spatial resolution, which refers to the discriminative power between neighboring nucleobases in a DNA strand. Strong optical confinement and consequently, high spatial resolution is one of the intrinsic properties of SERS. Although it has been demonstrated that adjacent molecules in dry conditions can be resolved by using tip-enhanced Raman spectroscopy[29], the high spatial resolution of SERS in aqueous media remains unverified. Here, we studied the spatial resolution of nanoslit SERS by measuring the spectroscopic fluctuations of a synthesized single-stranded DNA oligonucleotide sample, 5′-poly(dA)$_{48}$dCdG-3′. As nanoslit SERS is not fast enough to monitor the real-time translocations, we could only study the DNA strands temporally adsorbed inside the nanoslit (indicated in Fig. 4a). If the hot spot is small enough (e.g., sub-nanometer), the stochastic thermodynamic fluctuation of the DNA strand can randomly displace different bases into the hot spot generating SERS signals. We chose poly(dA)$_{48}$dCdG as the sample, because the triple-base of ACG at the end of the strand has a total length of ~1 nm, and the three bases have very distinct Raman bands in the range of 600–900 cm$^{-1}$. In Fig. 1b, we have learned that the SERS intensities of these nucleotides are comparable. The different affinities of A (stronger) and C, G (weaker) may provide a way to anchor the poly(dA) inside the pore but still allows stronger displacements of C and G in and out of hot spots. By studying the spectroscopic changes of the triple, we can evaluate the spatial resolution of nanoslit SERS. We prepared this DNA sample in a 10 mM KNO$_3$ solution with a low concentration of $1 \times 10^{-8}$ M and measured it by nanoslit SERS at a +0.3 V bias voltage.

Figure 4b shows a temporal contour map of SERS from poly(dA)$_{48}$dCdG inside the slit. The bias voltage is applied between 50 and 550 s. The bands of G, A, and C are marked by the green, blue, and orange frames, respectively. Compared to the spectroscopic library shown in Fig. 1b, the Raman bands of these three bases were slightly shifted. This may be related to both the influence from the backbone of DNA strands, the orientation of the

bases and the Stark effect induced by the high-electric field inside the pore[46]. Clearly, the intensity of the Raman band of A (~730 cm$^{-1}$) gradually increased and became relatively stable. The observation of continuous signals of A was important, as it indicated the temporally adsorption of the DNA strand at the hot spot.

In contrast to the continuous A signals, we can find many transient but asynchronous signals at the spectroscopic frequencies related to C and G. This blinking phenomenon suggests that only part of the DNA strand was measured. Or else we should observe strong A signals synchronized with weak C and G signals. In Supplementary Note 6, we further studied the FWHMs of these bands of A, G, and C. As shown in Supplementary Fig. 13, the FWHMs of C and G are ~10 cm$^{-1}$, while for A it is ~16 cm$^{-1}$. The results for G and C are consistent with the results from single-molecule sensing, while the A result corresponds to the ensemble measurement (e.g., Supplementary Fig. 2c). To better visualize the fluctuation of DNA sample, we plotted the time traces of the band intensities of A, C, and G in Fig. 4c. Clearly, the spikes of C and G are asynchronous. They are representing all permutation combinations of A, AC, AG, and ACG. The four exemplary spectra are also shown in Fig. 4c. As the length of three bases is ~1 nm, this asynchronous blinking strongly indicates that the nanoslit SERS has an extremely high spatial resolution at the sub-nanometer scale. The randomly appearances of C and G signals further indicate that the adsorbed DNA strand may follow a stochastic fluctuation process, rather than a creeping process on the gold edge of the slit. In Supplementary Note 7 and 8, we have repeated the experiments on both the same-sequence DNA sample and another DNA sample with a different sequence such as poly(dC)$_{28}$dGdA. As shown in Supplementary Fig. 14-16, we can also observe asynchronous blinks of nucleobases incorporated in these DNA strands. It should be noted, we also observed fast blinking events, mostly with signals from the dominant bases rather than all bases. These signals could be related to very short-time adsorptions of DNA strands. And these reproducible results can further support the high spatial resolution of nanoslit SERS.

## Discussion

In this work, we have studied the feasibility of spectroscopic nanopore sensing based on plasmonic nanoslit SERS. The combination with nanofluidics facilitates the capture and adsorption of analytes at the hot spots of the slit. By studying the DNA analytes adsorbed inside the nanoslit, we have established a spectroscopic library of four nucleotides for identification. These adsorbed molecules follow stochastic fluctuations, providing us a way to study the fundamental properties of nanoslit SERS in real-time sensing on the one hand. But on the other hand, the stochastic process brings challenges in sequential reads along the DNA strand. Although a plasmonic nanopore or nanoslit has a potential of manipulating nano-objects by the plasmonic gradient forces[30,47], measuring DNA translocations in real-time is tough for a standard optical technique. The translocation of DNA through a solid-state pore is still too fast for a spectrometer[37], and the different affinities of nucleotides may also introduce uncertainties. One may combine nanoslit SERS with advanced spectroscopic techniques, such as Fourier-transform coherent anti-Stokes Raman spectroscopy to open up new avenues toward ultrafast, real-time, and broadband spectroscopic sensing[48,49]. As a general advantage of being a kind of solid-state nanopore technologies, nanoslit SERS can become more practical though, when it is integrated with other established advanced technologies for ratcheting DNA[50], and for opto-electro dual-channel sensing[51]. Although we show critically important properties of nanoslit SERS for real-time single-molecule identification, the

remaining hurdles for nucleic acid sequencing are still high. For single-molecule studies though, nanofluidic SERS based on the described plasmonic nanoslit cavities or other similar plasmonic aperture structures, can provide a promising way for highly localized, label-free, and background-free sensing in real time.

## Methods

**Nanoslit chip fabrication.** The nanoslit devices were fabricated by standard micromachining processes[33]. In brief, the fabrication process started from a silicon-on-insulator wafer, and it used the standard electron beam/UV lithography, anisotropic Si wet etching (KOH based), and buffer HF etching to make the cavity, the gratings and the nanoslit (~50 nm) in a freestanding Si membrane (700 nm thick). The freestanding Si membrane was 50–100 μm in size, made by a deep etching process. Then we deposited an additional CVD coating of a 50 nm silicon oxide layer at the backside before Au sputtering, which can reduce the capacitance and noise[35]. Plasmonic nanoslits were formed after the Au sputtering. The gap size of nanoslits were shrunk below 10 nm, characterized by both SEM (Philips XL 30 at 5 kV) and TEM (Tecnai F30 at 300 kV). The thickness of the sputtered Au layer was adjusted between 100 and 200 nm. However, as shown in Supplementary Fig. 1d, the surface of gold was not super smooth, and nanoscaled knobs appeared everywhere, which could further shrink the gap size and improve the plasmonic enhancement for SERS.

The resultant nanoslit cavities had the following geometrical parameters. (1) Nanoslits had a length of ~1 μm, and a width of <10 nm. (2) Inverted prism cavities had a top length of ~2 μm, a top width of ~1 μm, a depth of 800–900 nm, and a fixed vertex angle of 70.5 degree. (3) The Bragg-mirror gratings had a pitch of 290 nm, a width of 100 nm, and a length of 3 μm. All parameters were optimized for 785 nm excitation in water. The fabricated nanoslit chip was 18 mm × 18 mm in size. The membrane was at the center of chip, with a single nanoslit cavity at its center. Currently, we have also successfully fabricated such nanoslit chips at the wafer-level at our in-house 200 mm CMOS pilot line (see Supplementary Fig. 1b).

**Materials selections.** Different to most nanopore sensing technologies based on resistive pulses, the spectroscopic readout of nanoslit SERS is much less sensitive to the electronic features of the device (e.g., low electronic noise), but more sensitive to the chemicals and materials which can influence SERS. We found halide electrolytes could strongly corrode gold and thus malfunction SERS. To solve this problem, we used $KNO_3$-based electrolyte solutions at a low concentration of 10 mM. The Raman background signals from this solution can be negligible. Passivation of the silicon membrane by a dielectric layer (see the red layer in Supplementary Fig. 1c) was another critical step. It could first reduce the capacitance of the membrane. Secondary, it could reduce the bipolar electrochemistry happening on the gold surface and meanwhile it interrupted the plasmons induced interactions at the silicon-water interface[52]. The reduction of side effects could make fast and reproducible fluidic responses modulated by the bias voltages. The electric circuit was used for applying the voltage and monitoring the possible fluidic problems during the measurement, such as failed wetting (GΩ resistance) and clogging (dramatically increased resistance). Due to the high background noises shown in our set-up, we currently cannot use the ionic current pulses to track DNA translocations. But we may solve it by passivating the entire Si layers[53].

Most chemicals were ordered from Sigma-Aldrich, including dGMP (2′-Deoxyguanosine 5′-monophosphate sodium salt hydrate >99%), dAMP (2′-Deoxyadenosine 5′-monophosphate, 98–100%), dTMP (Thymidine 5′-monophosphate disodium salt hydrate, >99%), and dCMP (2′-Deoxycytidine 5′-monophosphate sodium salt >98%), adenine (>99%), adenine-1,3-$^{15}N_2$ (>98%, and 98% $^{15}N$ atoms), $KNO_3$ (>99%), and $HNO_3$ (70%). The purity is provided by Sigma-Aldrich. We directly used them without any further purification. DNA oligonucleotides powders were ordered from Integrated DNA Technologies, Inc. The sample solutions were prepared in 10 mM $KNO_3$ solutions.

**Flow cell setup.** Plexiglas was used to make our custom-made flow cells. The transparency helped to check whether bubbles were trapped inside the bottom cavity. Supplementary Fig. 1 shows a 3D model of the flow cell. It had a top part and a bottom part. The top part was a truncated conical chamber, facilitating the approaching of the objective lens to the chip surface. The bottom chamber had a cylindrical chamber with a diameter of 10 mm and a depth of 2.5 mm. The inlet and outlet channels were connected on the sides. There was a separate channel for the Ag/AgCl electrode on each part, with a fix distance of 9 mm. The nanoslit chip separated the flow cell into two chambers. Freshly prepared Ag/AgCl electrodes were prepared and cleaned before use. All sealing rings and electrodes were disposable. An Axon Axopatch 200B (Molecular devices, LLC) was mainly used to supply a bias voltage cross the membrane for driving motions of analytes. Oxygen plasma process (5 min) was used to clean both topsides and bottom sides of all flow cells and nanoslit chips to avoid any possible contaminations. To completely wet the nanoslit, a droplet of mixture of isopropyl alcohol and water (1:1 in volume) was placed near the cavity, and the liquid can fill inside the slit through the Marangoni effect[54]. This was important to avoid trapping air bubbles inside the slits and thus can improve the reproducibility of experiments. Three times rinsing

by deionized water was necessary before loading the chip into the flow cell. The bias voltage was applied at the working electrode in the *trans*-chamber (on the backside of the nanoslit chip), while a counter electrode was placed in the *cis*-chamber. The two electrodes were spaced at 9 mm apart. A 10 mM $KNO_3$ solution was used as the electrolyte solution. The applied voltage was ranging between +0.3 and +1 V.

**Raman and SERS experiments.** We used a Raman setup (Witec alpha-300) equipped with a Front illuminated EMCCD (Andor, DU970N-FI) for the SERS measurements. A 785 nm laser XTRA (Toptica), a spectrometer with 600, and 1200 grooves/mm grating and a 60× water immersion lens (Olympus, LumplanFLN60x, NA = 1.0) were used in all nanoslit SERS experiments. The laser power was always at 8 mW delivered on samples, and each spectrum was acquired between 0.1 and 0.5 s. A Faraday metal cage was used for electromagnetic shielding, and it was also a darkroom for SERS measurements. All electronic facilities were connected to an uninterruptible power supply.

**Data processing.** To analyze data, we used Witec project software V2.10, Origin 8.5, and MATLAB R2016a scripts. (1) Smoothing and baseline correction. All spectra were filtered by using the Savitzky–Golay algorithm. The baseline was always subtracted in both the contour mapping and the peak fitting algorithms. In the former case, the baseline was subtracted by a polynomial equation which best fits the background of the spectrum. In the latter case, the baseline was subtracted locally near the interesting Raman bands, by defining it as a mean of 4 pixels near the start (710 cm$^{-1}$) and the end (750 cm$^{-1}$) of the band (e.g., 710–750 cm$^{-1}$).

(2) Spectroscopic features. For analyzing the spectroscopic features, namely peak wavenumber, FWHM, and intensity of Raman bands, we used two algorithms. One was the Gaussian fitting and the other was the local maximum fitting. In fitting of single Raman bands, they gave same results. However, for BiASERS experiment where it showed Raman bands with double or even multiple peaks (mix of adenines at $10^{-7}$ M), we used a combined algorithm. In Fig. 2b, we were only interested in the dominating bands, and we only used the local maximum fitting to directly find the strongest peaks from all 2500 bands. From the resultant dual-peak distribution of the peak wavenumbers, we can clearly observe the fluctuations of dominating adenine isotopologues. To further extract the FWHMs and peak wavenumbers of all bands including these weak and hidden ones, we must use a deconvolution algorithm. We first used the local max fitting to define the peak wavenumber and number of Raman bands in the spectroscopic range of 710–750 cm$^{-1}$. It may find one or several peaks in a spectrum. Then we used the Gaussian fitting to deconvolute these Raman bands in all 2500 spectra, and obtained the peak wavenumbers and FWHMs of all fitted 4410 Raman bands, representing different isotopic variations.

(3) Band intensity. The intensity of the Raman bands was integrated in its spectroscopic range, with the subtraction of baselines. For an example, the band intensity of A in Fig. 4c was integrated from 710 to 750 cm$^{-1}$, discarding the baseline.

To further support our conclusions, more experimental results and data analysis are available in the Supplementary Information.

**Data availability.** The authors declare that the main data supporting the findings of this study are available within the article and its Supplementary Information files. Extra data are available from the corresponding author upon request.

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

## Acknowledgements

C.C., P.N., S.C., and K.W. gratefully acknowledge financial supports from the FWO (Flanders). We also would like to thank Prof. Steven Chu from Stanford University for useful discussions.

## Author contributions

C.C., T.S., and P.V.D. designed the experiments, C.C., Y.L., P.N., and S.C. fabricated the devices, C.C., Y.L., and K.W. characterized the devices, C.C., Y.L., and S.K. performed experiments, C.C., Y.L., and P.N. analyzed results, and C.C., Y. L., T.S., and P.V.D. co-wrote the manuscript. All authors discussed the results and were involved in the manuscript preparation.

## Additional information

**Competing interests:** The authors declare no competing interests.

