## [Peer Review File · Nature Communications]

Editorial Note: This manuscript has been previously reviewed at another journal that is not operating a transparent peer review scheme. This document only contains reviewer comments and rebuttal letters for versions considered at Nature Communications. Mentions of prior referee reports have been redacted.

PEER REVIEW FILE

Reviewers' comments:

Reviewer #1 (Remarks to the Author):

This much-improved manuscript by Chen and co-workers describes an experimental study aimed at investigating the feasibility of DNA sequencing using nanopore surfaces-enhanced Raman spectroscopy (SERS). Over the past rounds of review, the authors have implemented a number of suggestions with regard to the presentation of the material, analysis of experimental data and performed several additional measurements. As detailed in the Authors' reply letter, not all experiments produced the expected outcome. At this point of review, it seems unlikely that a set of additional measurements would substantially improve the quality of the data and hence the strengths and shortcomings of the study are evaluated below from the material presented.

Strengths:

The Authors reproduced previous bulk measurement of SERS spectra using a voltage-loaded nanofluidic system, showing that high concentration solutions containing purely A, C, G and T mononucleotides have distinct SERS spectra.

At a lower concentration of mononucleotides, the nanofluidic system was shown to be capable of detecting adsorption of single mononucleotides to the plasmonic hotspot.

The system is also shown to have spatial resolution that permits detection of single C and G nucleotides from the background of a 48-nucleotide polyA strand. The Authors were able to reproduce their data in an independent set of measurements.

Shortcomings:

The system considered by the authors is not a nanopore SERS system, it is a nanogap SERS system. The nanogap is 1 micron wide and contains multiple hotspots. The signals reported by the Authors very likely originate from multiple hot spots distributed along the gap, which is a key difference in comparison to the SERS nanopore system, where one or two hot spots positioned at the aperture of the nanopore, ensuring that electric field-driven transport loads charged molecules to the same hotspots. The Authors do not acknowledge this key difference in the manuscript neither do they discuss the possibility of the SERS signal coming from multiple hotspots along the nanogap.

The newly added analysis of the Raman traces and the Markov model finally show detection of single molecule. According to Table S2, the truly single molecule event constitute only 0.7% of all events (clear single peaks events such as presented in Figure 2A. The rest of events are either “few molecule events” (asymmetrically distributed peaks) or “many molecule events” (single wide peak). Whereas there is nothing wrong with having only 1% of all traces corresponding to truly single molecule events, the low occurrence of such events must be explicitly acknowledged in the main text of the manuscript.

According to the authors' reply letter, the system described by the authors works only for one specific combination of the parameters: 0.5V bias, 0.1s acquisition time and 10^{-7} nucleotide concentration. Apparently any deviation from the protocol (lowering bias, changing acquisition time or nucleotide concentration) does not produce the expected change in the signal. These negative results are not mentioned in the manuscript text.

In summary, the revised manuscript presents very interesting set of experiments that characterized fidelity of a nanogap SERS system for single molecule detection of DNA. In the opinion of this reviewer, the manuscript may be suitable for publication after implementing the following changes to the main text of the manuscript:

Replace “nanopore SERS” with “nanogap SERS” everywhere in the manuscript, including the title.

Discuss the difference between nanopore SERS and nanogap SERS in the introduction.

Acknowledge the very high possibility of having multiple hotspots along the nanogap contributing to the SERS signal.

Explicitly state the low occurrence of single molecule events (<1%) in the mononucleotide SERS

data.

Explicitly state the very high sensitivity of the measurement system with regard to the experimental conditions, such as voltage bias, signal acquisition time and nucleotide concentrations.

Explicitly state the range of voltages, concentrations and acquisition times explored and acknowledge the negative results obtained.

Replication of poly(dA)_46CG data (Figure R4) must be added to SI, it will strengthen the manuscript considerably.

Reviewer #2 (Remarks to the Author):

I believe that the authors have again made a credible attempt to respond to reviewer concerns. However, I am not able to evaluate many of the technical details of their response. My impression is that the response is useful but not compelling. As in response to the previous review, there is focus on reproducibility of results.

Nevertheless, in my opinion, experiments and rationales are well-enough documented that readers will be able to either extend these results in their own experiments, or to discern flaws in the reasoning or results reported here, and thus I would not object to publication of the article. Progress is needed on this technology, and it may be the case that publication will help to spur on the thinking of others in the field -- to pick up and extend the good ideas, and perhaps to develop alternatives to, or clearly state the flaws in, the less-good ideas.

But my recommendation for publication is conditional on further qualification of statements about conclusions that can be drawn from the research. The fact is that the authors provide no evidence for single-molecule sensitivity for 2 of the 4 nucleotides, and only partial evidence for a third nucleotide (G) (and reviewer 1 stated that "SM detection of Adenine had been demonstrated in previous studies") so perhaps stated conclusions about single-molecule sensitivity need to be further qualified. It appears to me to be a partial step forward that they present evidence for single-molecule sensing of G. While it is interesting that they state in the rebuttal their opinion about the feasibility in principle to detect other bases at single molecule level, an opinion without evidence is generally not an adequate basis for publishing a research report. There already exists reasonable literature on the theoretical potential for SERS for DNA sequencing, so this group's conviction that single-molecule SERS for other-than-A is possible, does not represent a significant advance. Here, I defer to the other reviewers' evaluation of the

strength of the new evidence for single-molecule sensitivity for G. If evidence for single-molecule sensitivity remains equivocal, then this calls into question the usefulness of the report, because, as the authors say, this is one of three conditions required for nanopore SERS to ever be a potentially useful DNA sequencing method.

Reviewer 3 called out a particularly egregious overstatement. To the authors' credit, some statements are better qualified throughout the manuscript (beginning in the abstract). But this needs more work. For example, instead of "REVEAL THE...single-molecule sensitivity and subnanometer spatial resolution of nanopore SERS, ESTABLISH ITS ABILITY for the single-molecule study", one might instead say, "PROVIDE EVIDENCE FOR." Further, "intrinsic and unique seem to me to be vague and over-claiming. It has already been established that electronic detection of DNA sequence using nanopores provides single-molecule sensitivity and subnanometer spatial resolution, so these observations in a nanopore detector are not unique for SERS. Similarly, in the summary paragraph, "provide evidence for" should probably replace "demonstrate."

The other thing that seems missing to me, is a motivation for SERS as an alternative to electronic detection for DNA sequencing with nanopores. What advantages would SERS convey? This point was raised in a previous review but not addressed.

In conclusion, I think it is important for experimental results such as these to get into the literature. It is also important for the credibility of the authors and the journal (and the reviewers) that the article not overstate the conclusions that can be drawn from those experiments.

Reply letter

Again, we really appreciate all comments from reviewers. All these constructive and critical comments during the three reviewing rounds did help us to make our work more solid and relevant. In summary,

1. We agree with the reviewer 1, “nanopore SERS” would not be very suited for describing a 10x1000 nm aperture. We propose to change it to “nanoslit SERS”, a term which we have also used in our previous publications.
2. In this revised manuscript, we have further toned down our claims as suggested, explicitly stated the pros and cons of nanoslit SERS, and added discussions on our other unshown but tested experimental results.
3. Although we consider our single-molecule sensing of other nucleotides like dGMP is solid, to further convince reviewer 2, we have added a new experimental result (Fig. S12) of asynchronous blinking (step-like) of dCMP in BiASERS of a mixture of all four nucleotides. We didn't show this result before, as we believe the approach of using isotopologues is the most solid method to demonstrate single-molecule sensitivity. Still, we can employ nanoslit SERS for detecting other nucleotides at the single-molecule level, even given the additional complexity by their different affinities to the gold surface.

Reply to reviewers' comments:

Reviewer #1 (Remarks to the Author):

This much-improved manuscript by Chen and co-workers describes an experimental study aimed at investigating the feasibility of DNA sequencing using nanopore surfaces-enhanced Raman spectroscopy (SERS). Over the past rounds of review, the authors have implemented a number of suggestions with regard to the presentation of the material, analysis of experimental data and performed several additional measurements. As detailed in the Authors' reply letter, not all experiments produced the expected outcome. At this point of review, it seems unlikely that a set of additional measurements would substantially improve the quality of the data and hence the strengths and shortcomings of the study are evaluated below from the material presented.

Strengths:

The Authors reproduced previous bulk measurement of SERS spectra using a voltage-loaded nanofluidic system, showing that high concentration solutions containing purely A, C, G and T mononucleotides have distinct SERS spectra.

At a lower concentration of mononucleotides, the nanofluidic system was shown to be capable of detecting adsorption of single mononucleotides to the plasmonic hotspot.

The system is also shown to have spatial resolution that permits detection of single C and G nucleotides from the background of a 48-nucleotide polyA strand. The Authors were able to reproduce their data in an independent set of measurements.

Shortcomings:

The system considered by the authors is not a nanopore SERS system, it is a nanogap SERS system. The nanogap is 1 micron wide and contains multiple hotspots. The signals reported by the Authors very likely originate from multiple hot spots distributed along the gap, which is a key difference in comparison to the SERS nanopore system, where one or two hot spots positioned at the aperture of the nanopore, ensuring that electric field-driven transport loads charged molecules to the same hotspots. The Authors do not acknowledge this key difference in the manuscript neither do they discuss the possibility of the SERS signal coming from multiple hotspots along the nanogap.

The newly added analysis of the Raman traces and the Markov model finally show detection of single molecule. According to Table S2, the truly single molecule event constitute only 0.7% of all events (clear single peaks events such as presented in Figure 2A. The rest of events are either “few molecule events” (asymmetrically distributed peaks) or “many molecule events” (single wide peak). Whereas there is nothing wrong with having only 1% of all traces corresponding to truly single molecule events, the low occurrence of such events must be explicitly acknowledged in the **main text** of the manuscript.

According to the authors' reply letter, the system described by the authors works only for one specific combination of the parameters: 0.5V bias, 0.1s acquisition time and 10^{-7} nucleotide concentration. Apparently any deviation from the protocol (lowering bias, changing acquisition time or nucleotide concentration) does not produce the expected change in the signal. These negative results are not mentioned in the manuscript text.

In summary, the revised manuscript presents very interesting set of experiments that characterized fidelity of a nanogap SERS system for single molecule detection of DNA. In the opinion of this reviewer, the manuscript may be suitable for publication after implementing the following changes to the main text of the manuscript:

Replace “nanopore SERS” with “nanogap SERS” everywhere in the manuscript, including the title. Discuss the difference between nanopore SERS and nanogap SERS in the introduction. Acknowledge the very high possibility of having multiple hotspots along the nanogap contributing to the SERS signal.

We have changed “nanopore SERS” to “nanoslit SERS” in the entire text. And we have also added a discussion on the difference between a nanoslit and a nanopore in page 3.

Explicitly state the low occurrence of single molecule events (<1%) in the mononucleotide SERS data.

We have added a discussion on this part in page 4 and 9. The low single-molecule events is related to the fact our BiASERS is a very special case, compared to the conventional BiASERS.

Nowadays, single-molecule SERS experiments have been reported by many independent groups, especially when using the BiASERS strategy. The key is the spatially random absorption of molecules at many individual hot spots of a SERS substrate (e.g. rough metal surface) or dynamic formation or passing of SERS substrates (e.g. assembled nanoparticles) with molecules at the sensing region. In the temporal domain, different combinations of molecules and hot spots alternatively appear inside the sensing region. However, in our nanoslit SERS, the hot spots were spatially and temporally fixed. It means that the in-site spatial fluctuation (e.g. stochastic absorption, desorption, or rotation) of molecules at the

hot spots is mandatory for observing single-molecule events. We did put considerable efforts to screen the setting to bring an appropriate stochastic process into the system. This makes our single-molecule experiments highly sensitive to the measurement setting. And we observed much less true single-molecule events than the conventional BiASERS.

*Explicitly state the very high sensitivity of the measurement system with regard to the experimental conditions, such as voltage bias, signal acquisition time and nucleotide concentrations.
Explicitly state the range of voltages, concentrations and acquisition times explored and acknowledge the negative results obtained.*

These two points have been discussed in page 10.

Replication of poly(dA)_46CG data (Figure R4) must be added to SI, it will strengthen the manuscript considerably.

We have moved this result to SI Figure S15.

Reviewer #2 (Remarks to the Author):

I believe that the authors have again made a credible attempt to respond to reviewer concerns. However, I am not able to evaluate many of the technical details of their response. My impression is that the response is useful but not compelling. As in response to the previous review, there is focus on reproducibility of results.

Nevertheless, in my opinion, experiments and rationales are well-enough documented that readers will be able to either extend these results in their own experiments, or to discern flaws in the reasoning or results reported here, and thus I would not object to publication of the article. Progress is needed on this technology, and it may be the case that publication will help to spur on the thinking of others in the field – to pick up and extend the good ideas, and perhaps to develop alternatives to, or clearly state the flaws in, the less-good ideas.

But my recommendation for publication is conditional on further qualification of statements about conclusions that can be drawn from the research. The fact is that the authors provide no evidence for single-molecule sensitivity for 2 of the 4 nucleotides, and only partial evidence for a third nucleotide (G) (and reviewer 1 stated that "SM detection of Adenine had been demonstrated in previous studies") so perhaps stated conclusions about single-molecule sensitivity need to be further qualified. It appears to me to be a partial step forward that they present evidence for single-molecule sensing of G. While it is interesting that they state in the rebuttal their opinion about the feasibility in principle to detect other bases at single molecule level, an opinion without evidence is generally not an adequate basis for publishing a research report. There already exists reasonable literature on the theoretical potential for SERS for DNA sequencing, so this group's conviction that single-molecule SERS for other-than-A is possible, does not represent a significant advance. Here, I defer to the other reviewers' evaluation of the strength of the new evidence for single-molecule sensitivity for G. If evidence for single-molecule sensitivity remains equivocal, then this calls into question the usefulness of the report, because, as the authors say, this is one of three conditions required for nanopore SERS to ever be a potentially useful DNA sequencing method.

In single-molecule experiment, using 2 of the 4 nucleotides is interesting, but it is not as convincing as using a pair of isotopic nucleotides. We have started from using 2 or even more of the 4 nucleotides in BiASERS (e.g. A+C, A+G, C+G, A+C+G, A+C+G+T...). In most time, we observed dominated signals from 1 of the 2 or 3 or even 4 nucleotides. Sometimes, we did also observe few continuous blinking events of other nucleotides. An example is shown in Fig. R1 below. In the mixed solution of four nucleotides, we observed stronger blinking signals of dCMP with relative stable signals of dGMP, though we did not observe signals from the other two nucleotides. The step-like fluctuations and the narrower FWHM from dCMP indicate that other nucleotides rather than adenine can also be detected at the single-molecule level. But, we should also mention that using chemically different molecules will reduce the reproducibility and the probability of asynchronous blinking events.

To study single-molecule sensitivity of our nanoslit SERS, we need to record stochastic fluctuations of temporally adsorbed molecules. In our opinion, the use of isotopologues is so far, the best way to eliminate the influence of different affinities of analytes.

In answer to the reviewer's comment, we have added the result below into the SI Figure S15, and discussed the result in the main content (page 10) too.

Reviewer 3 called out a particularly egregious overstatement. To the authors' credit, some statements are better qualified throughout the manuscript (beginning in the abstract). But this needs more work. For example, instead of "REVEAL THE...single-molecule sensitivity and sub-nanometer spatial resolution of nanopore SERS, ESTABLISH ITS ABILITY for the single-molecule study", one might instead say, "PROVIDE EVIDENCE FOR." Further, "intrinsic and unique seem to me to be vague and over-claiming. It has already been established that electronic detection of DNA sequence using nanopores provides single-molecule sensitivity and subnanometer spatial resolution, so these observations in a nanopore detector are not unique for SERS. Similarly, in the summary paragraph, "provide evidence for" should probably replace "demonstrate."

As far as we know, biological nanopore can indeed electrically read and identify bases for sequencing, but this has not been shown yet for solid-state nanopores. However, we still would like to further tone down our claims and add some discussions on our contribution to the advancements of solid-state nanopores.

We have toned down our statements in the whole text accordingly.

The other thing that seems missing to me, is a motivation for SERS as an alternative to electronic detection for DNA sequencing with nanopores. What advantages would SERS convey? This point was raised in a previous review but not addressed.

We do not intend to claim SERS can replace current electric sensing. It's too early to discuss nanoslit/nanopore SERS for real sequencing applications. With the current sampling speed, nanoslit/nanopore SERS is difficult to be a practical or even a complementary method for nanopore sequencing.

Our initial motivation is that (1) SERS can identify a molecule directly from its spectroscopic information; (2) spectroscopic detection won't interfere with electronic controls or detection; and (3) plasmonic force may slow down DNA motions. With our current result on the sub-nanometer spatial resolution, we now have a new motivation that SERS is ideal for measuring small molecules like DNA.

In addition, the large-mass fabrication of nanoslit SERS is feasible (as shown in Fig. S1). To identify single nucleotides, a sub-10 nm slit works well. This size is much more achievable and scalable by current CMOS processing than other kind of solid-state nanopores which require sub-2 nm for identification.

Of course, there are also downsides for our nanoslit SERS approach. (1) The single-molecule SERS measurements are too slow. We have discussed the possibilities to solve it in the text, but it requires much further efforts. (2) Massive parallelization of SERS spectrometers is also difficult. Miniaturization of spectrometers would be useful for solving this problem, such as developing on-chip Raman setup has been started by several groups, including us. The success of this kind of work may bring multi-channel SERS sensing in future. Alternatively, we may only monitor certain Raman bands (e.g. 600-900 cm^{-1}), rather than the

whole spectral range. This may facilitate the scalability and reading speed of nanoslit/nanopore SERS.

We have added more discussions in the revised text.

In conclusion, I think it is important for experimental results such as these to get into the literature. It is also important for the credibility of the authors and the journal (and the reviewers) that the article not overstate the conclusions that can be drawn from those experiments.

Reviewers' Comments:

Reviewer #1 (Remarks to the Author):

The authors have adequately addressed all comments of this reviewer

Reviewer #2 (Remarks to the Author):

The authors have made a number of interesting and important observations on a path toward a solid-state device for DNA analysis using SERS signals. Over the course of three submissions/revisions, the manuscript has been significantly improved by additional experiments and by including a more balanced presentation of the strengths and challenges of the system and measurements it enables. The authors have consistently demonstrated attention to reproducibility. As such, in the opinion of this reviewer, the manuscript now presents a substantive contribution to the field and is worthy of publication.

Jeffery A. Schloss